# Identification of Circular RNAs of Testis and Caput Epididymis and Prediction of Their Potential Functional Roles in Donkeys

**DOI:** 10.3390/genes14010066

**Published:** 2022-12-25

**Authors:** Yan Sun, Yonghui Wang, Yuhua Li, Faheem Akhtar, Changfa Wang, Qin Zhang

**Affiliations:** 1Shandong Provincial Key Laboratory of Animal Biotechnology and Disease Control and Prevention, College of Animal Science and Technology, Shandong Agricultural University, Taian 271018, China; 2Liaocheng Research Institute of Donkey High-Efficiency Breeding and Ecological Feeding, Liaocheng University, Liaocheng 252059, China

**Keywords:** circRNAs, testis, caput epididymis, donkey

## Abstract

Circular RNAs (circRNAs) are a class of noncoding RNAs with a covalently closed loop. Studies have demonstrated that circRNA can function as microRNA (miRNA) sponges or competing endogenous RNAs. Although circRNA has been explored in some species and tissues, the genetic basis of testis development and spermatogenesis in donkeys remain unknown. We performed RNA-seq to detect circRNA expression profiles of adult donkey testes. Length distribution and other characteristics were shown a total of 1971 circRNAs were differentially expressed and 12,648 and 6261 circRNAs were detected from the testis and caput epididymis, respectively. Among these circRNAs, 1472 circRNAs were downregulated and 499 circRNAs were upregulated in the testis. Moreover, KEGG pathway analyses and Gene Ontology were performed for host genes of circRNAs. A total of 39 upregulated circRNA host genes were annotated in spermatogenesis terms, including *PIWIL2*, *CATSPERD*, *CATSPERB*, *SPATA6*, and *SYCP1*. Other host genes were annotated in the focal adhesion, Rap1 signaling pathway. Downregulated expressed circRNA host genes participated in the TGF-β signaling pathway, GnRH signaling pathway, estrogen signaling pathway, and calcium signaling pathway. Our discoveries provide a solid foundation for identifying and characterizing critical circRNAs involved in testis development or spermatogenesis.

## 1. Introduction

Donkeys (*Equus asinus*) belong to the equine family. Thousands of years ago, donkeys were used for transporting goods and facilitated communications in distant areas. The donkey industry has produced more economic benefits by exploiting donkey skin, milk, and meat. It is essential for carrying out the breeding of donkeys because of plummeting stocks. Jackass fertility plays a decisive role in donkey reproduction. Besides the requirements of morphological assessment, testicular development and sperm quality are the most critical indicators. In the early stages, we investigated the mRNAs and microRNAs (miRNAs) expression of testis and cauda epididymis in jackass. Noncoding RNAs (ncRNAs), such as miRNAs, Piwi-interacting RNAs (piRNAs), and long noncoding RNAs (lncRNAs), are vital regulators for gene expression. The essential genes, regulated events, and critical signaling pathways related to reproductions were summarized [1]. In recent years, circular RNAs (circRNAs) have ubiquitously been in covalently closed-loop form in mammalian cells as an important class of post-transcriptional regulators [2]. CircRNAs presented significant tissue and developmental stage specificity; their primary role is modulating microRNAs as sponges to protect their mRNA targets from degradation [3]. Therefore, circRNAs are known as part of the competitive endogenous RNA network (ceRNET). Besides tethering activity, circRNAs also regulate the transcription of their host genes [4] and the splicing of their cognate mRNA through R-loop formation [5]. Recent studies suggested that circRNAs act as microRNA sponges and play an important role in regulating gene expression through a circRNA-microRNA-gene pathway [2,6].

CircRNAs involved in reproductive development have been reported. The well-known circRNA related to reproductive processes was from the sex-determining region Y (*SRY*) gene and performed specific functions in adult mouse testis [7]. The number of circRNAs in the testis rank second to the amount found in rats’ brains [8,9]. The expression profiles of circRNAs of human testis tissue were delineated using high-throughput sequencing, and it was discovered that circRNA-forming genes are mostly related to spermatogenesis and fertilization [10]. Liang et al. constructed a circRNA database of pigs [11]; in the testis tissue, circRNAs showed the most incredible abundance and the most significant number of specific circRNAs. Gao et al. detected circRNAs expression in neonatal and adult Chinese Qinchuan cattle testes and performed functional analysis for the host genes of circRNAs [12]. The comparative analysis of circRNAs from three livestock species (pig, sheep, and cattle) suggested that the number of exonic circRNAs in the testis seems associated with the sexual maturity of the animal [13]. However, no further studies discussed the regulatory function of these circRNAs in mammalian testis development and spermatogenesis. Our understanding of circRNAs, their production, and their function remains limited. 

We assembled one male Dezhou donkey genome, the first chromosome-level donkey reference genome [14]. This reference and annotations version is more helpful for analyzing the role of regulators. Therefore, in this study, to further explore the complicated regulatory network of circRNAs, microRNA, and protein-coding genes in donkey testis and caput epididymis, high-throughput sequencing and bioinformatics analyses were performed. The general molecular biological characteristics of circRNAs were also determined. A circRNA-miRNA-mRNA network related to reproduction was subsequently constructed to explore the interaction among different molecules.

## 2. Materials and Methods

### 2.1. Sample Preparation, Ethics Statement

The Institutional Animal Care and Use Ethics Committee approved the animal experiments and study protocol of Shandong Agricultural University (SDAUA-2018-018). Normal and intact donkey testis tissues were collected from male Dezhou donkeys. Three adult donkeys were derived from Yucheng Huimin Agricultural Science and Technology Co. Ltd (Yucheng, Shandong, China). The testis and caput epididymidis samples were sliced and immediately preserved in liquid nitrogen until use. Finally, tissue samples were sent to the Beijing Omics Biotechnology Co. LTD for library construction and sequencing.

### 2.2. RNA Isolation, Library Construction, and Illumina Sequencing

The total RNA from six samples (the testis and caput epididymidis, each with 3 samples) was extracted using Trizol reagent (Invitrogen, Carlsbad, CA, USA). Then, ribosomal RNA (rRNA) was removed from the DNA-free RNA using the Ribo-Zero™ Kit (Epicentre, Madison, WI, USA) according to the manufacturer’s instructions. RNase R removed linear RNA molecules. RNA integrity and quality were evaluated using the Agilent 2100 Bioanalyzer (Agilent Technologies, Santa Clara, CA, USA) and the NanoDrop ND-2000 Spectrophotometer (NanoDrop, Wilmington, DE, USA). 

Total RNA was randomly fragmented into small pieces and purified into fragments with 200–600 bp lengths. Cleaved RNA fragments were subsequently reverse-transcribed to create the final library, whose sequencing data will eventually identify mRNAs and circRNAs. For the identification of microRNAs, the library construction was performed using a small RNA Sample Prep Kit (TruSeq, Illumina). After detecting the insert size of the library, sequencing was completed using the Illumina NovaSeq 6000 platform.

The sequencing data in Fastq format were uploaded to the Sequence Read Archive under Accession Number PRJNA869196.

### 2.3. Read Quality Control and Mapping

High-quality clean data were generated from raw data by removing adapter sequences, duplicated sequences, and sequences that showed more than 10% unidentified nucleotides and more than 50% low quality (Quality score ≤ 10). Read quality was assessed through the FastQC program (v0.11.9). BWA software (v0.7.17) [15] was used to map the clean data to the Dezhou donkey reference genome, a chromosome-level reference genome [14]. 

### 2.4. Identification of circRNAs and microRNAs

After the clean reads were aligned to the reference genome, CIRI software [16] was applied to identify backsplice junction reads using the BWA-MEM algorithm [17] for circRNA prediction. The types of circRNA are classified according to their origin. The identified circRNAs were subjected to statistical analysis of chromosome distribution and length distribution. The expression of circRNAs was quantified by mapped backsplicing junction reads per million mapped reads (TPMs). The final criterion was the circRNAs were expressed at least in two biological replicates. 

The analysis steps of sRNA sequencing data for microRNA identification were described in a previous report [1]. MicroRNAs were identified by using unique reads blasting to the microRNA database (Release 22.1) and predicting the microRNAs by mirdeep2 software (v0.1.3) according to microRNA homology and high conservation [18].

### 2.5. Differentially Expressed RNA Identification and Pathway Analysis

Differentially expressed microRNA (DEM) and circRNA (DEC) identification between testis and caput epididymis was performed using the DESeq2, with fold change ≥1.5 and *p* < 0.05 as evaluation criteria. The differences in gene expression levels and statistical significance can be quickly detected through a volcano plot. RNAhybrid [19] and miRanda [20] software were used to predict the target genes of DEMs. We conducted functional enrichment analysis of host genes to study the main functions of circRNAs. To obtain a functional annotation, Gene Ontology (GO) terms and Kyoto Encyclopedia of Genes and Genomes (KEGG) were performed for host genes of circRNAs. Finally, a corrected *p*-value < 0.05 was the threshold for statistically significant correlation.

### 2.6. DEC and Target microRNA Interaction

Integrating ncRNA and mRNA expression may identify the functional links between ncRNAs and their target genes. Any RNA transcripts (circRNAs, lncRNAs, and mRNAs) with miRNA binding sites can compete for and mutually affect the same miRNA; these species are competing for endogenous RNAs [21]. The putative target microRNAs of DECs were performed using the miRanda (v 3.3a) software and TARGETSCAN (V7.2).

### 2.7. Quantitative Real-Time PCR Validation

To confirm individual circRNA, after incubation with RNase R (Epicentre), first-strand cDNA was synthesized with a random primer transcribed using the FastKing RT Kit (with gDNase) (Tiangen, China) according to the manufacturer’s protocol. qPCR was performed using the TB Green PCR Kit (TaKaRa, Dalian, China) on the LightCycler/LightCycler 480 System (Roche Diagnostics), according to the manufacturer’s instructions. The primers of circRNAs were outward-facing, which primed divergently and amplified only circularly, but not linear RNA [22]. Each 20 µL real-time RT-PCR reaction included 10 µL TB Green Premix Ex Taq Ⅱ (Tli RNaseH Plus, 2×), 2 µL cDNAs (100 ng) and 0.8 µL primers. PCR conditions consisted of one cycle at 95 °C for 30 s, followed by 40 cycles at 95 °C for 5 s and 60 °C for 20 s, with fluorescence acquisition at 95 °C for 5 s, 60 °C for 1 min, 95 °C for 15 s, and 50 °C for 30 s. Relative expression level was determined using the 2^−ΔΔct^ method with β-actin as the control. Finally, the relative expression results were compared with the RNA-seq data.

## 3. Results

### 3.1. General Characteristics of circRNA in Donkey Testis and Caput Epididymis

The average clean data of transcriptome sequencing was 10,276,318,400 bp, and the high-quality rate (Q20) was not lower than 97% in each sample (Appendix A). We identified one circRNA containing at least two backspliced junction reads. Based on their location on the genome, all circRNAs were widely scattered on all chromosomes and mainly distributed on chromosomes 2 and 3 (Figure 1A). The host genes of these circRNAs were derived from exonic regions (16232, 82.3%), intronic regions (2114, 10.7%), and intergenic regions (1345, 7.0%) (Appendix A). For exonic circRNAs, 61.69% circRNAs were composed of no more than 10 exons. The lengths were widely distributed, and the most exonic circRNAs (62.4%) were no more than 20 kb nucleotides (nt) (Figure 1B). A total of 12,648 and 6261 circRNAs were identified from the testis and caput epididymis, respectively. We found that 3928 circRNAs were shared by testis and caput epididymis, and circRNA abundance was quantified as TPM (Appendix A). According to the expression profiles of the circRNAs, the correlation coefficients among three testis samples were >0.97, and Pearson’s correlation coefficients of three caput epididymides were >0.88, indicating high relativity between samples (Appendix A). 

### 3.2. Feature and Function Analysis of circRNAs and Their Host Gene 

All circRNAs were mapped to 5102 corresponding host genes in donkey testis and caput epididymis. Of these genes, 40.5% produced only one circRNA, and 59.5% generated more than two circRNAs (Appendix A). The circRNAs expressed in the testis were derived from 3904 host genes, and circRNAs expressed in caput epididymis were derived from 2802 host genes. We conducted KEGG annotation for the coexpressed and specifically expressed host genes in testis and caput epididymis, a total of 202 pathway terms, showing a high level of significance (corrected *p*-value < 0.05) including focal adhesion (Ko04510), oocyte meiosis and maturation (Ko04114, Ko04914), GnRH signaling pathway (Ko04912), and estrogen and calcium signaling pathway (Ko04915 and Ko04020) (Appendix A). 

### 3.3. Feature and Function Analysis of microRNAs in Donkey Testis and Caput Epididymis

For small RNA sequencing, the average number of unique reads was 13,825,528 bp after conducting secondary quality control in each sample (Appendix A). Finally, we identified 647 microRNAs; among them, 578 microRNAs were expressed in testis, and 511 microRNAs were expressed in caput epididymidis. There were 135 microRNAs specifically expressed in testis, and 68 microRNAs specifically expressed in caput epididymidis (Appendix A).

We identified 232 DEMs between testis and caput epididymidis, including 137 upregulated microRNAs and 95 downregulated microRNAs (Appendix A). The DEM result was also presented using a volcano plot (Figure 2A). A total of 21,892 target genes of DEMs were predicted; the result is shown in Appendix A.

### 3.4. Analysis of Differentially Expressed circRNAs between Testis and Caput Epididymis

According to the circRNA expression profiles, 1971 circRNAs were differentially expressed (Appendix A). Compared with caput epididymis, 1472 DECs were downregulated, and 499 DECs were upregulated in the testis (Figure 2B). To some extent, the function of circRNAs is reflected through their host genes. We conducted functional pathway and Gene Ontology analysis on the host genes. A total of 307 GO terms significantly enriched on host genes of upregulated expressed circRNAs are shown in Appendix A. The prominent GO terms include germ cell development (GO:0007281), male germ cell nucleus (GO:0001673), and sperm-egg recognition (GO:0035036). The result showed that 39 host genes were annotated in spermatogenesis terms (GO:0007283), including essential genes associated with reproduction of piwi-like RNA-mediated gene silencing 2 (*PIWIL2*), cation channel sperm-associated auxiliary subunit delta (*CATSPERD*), cation channel sperm-associated auxiliary subunit β (*CATSPERB*), spermatogenesis-associated 6 (*SPATA6*), and sperm flagellar 2 (*SPEF2*). In total, 298 circRNAs were from these host genes. Moreover, boule homolog, RNA-binding protein (*BOLL*), and synaptonemal complex protein 1 (*SYCP1*) produced 22 circRNAs. Pathway analysis can provide more information about the biological functions of genes. We enriched 77 significantly clustering functional pathways (Appendix A), including oocyte maturation and meiosis (ko04914 and ko04114), focal adhesion (ko04510), and Rap1 signaling pathway (ko04015).

For host genes of downregulated expressed circRNAs, we predicted 72 significantly enriched signal transduction pathways (Appendix A), including the TGF-β signaling pathway (ko04350), GnRH signaling pathway (ko04912) and estrogen signaling pathway (ko04915), calcium signaling pathway (ko04020), and oocyte meiosis (ko04114). A total of 22 host genes were annotated in the above pathways, including inositol 1,4,5-trisphosphate receptor type 2 (*ITPR2*), ATPase plasma membrane Ca ^2+^ transporting (*ATP2B1*), calcium/calmodulin-dependent protein kinase II, delta (*CAMK2D*), and epidermal growth factor receptor (*EGFR*). The GO annotation indicated that the host genes were involved in many biological processes (Appendix A), such as protein binding, signal transduction, and metabolic process, in which genes *ITPR2* and *ATP2B1* were annotated in calcium ion transmembrane transporter activity (GO:0015085).

### 3.5. The Target microRNAs of Differentially Expressed circRNAs in Donkey Testis and Caput Epididymis

The resulting circRNA-microRNA association network provided nodes and connections between circRNAs and their target microRNAs. Finally, 229 target microRNAs of DECs were predicted (Appendix A). According to our data, one circRNA may show tethering activity toward multiple microRNAs as targets. For example, 22 target microRNAs were predicted for 7 upregulated expressed circRNAs from the *PIWIL2* host gene. On the other hand, more than two circRNAs show tethering activity toward the same miRNA as targets. eca-miR-205 was the target microRNA of multiple DE_circRNAs from host genes *EGFR*, *CATSPERD*, *ATP2B1*, *PIWIL2*, and the microRNA Eca-miR-324-5P was the target of circRNAs from genes *CATSPERB*, *SPEF2*, *CAMK2D*, and *PIWIL2*. Eca-miR-100 was the target of circRNA from genes *CATSPERD*, *ATP2B1*, and *CATSPERB*.

### 3.6. Experimental Validation of Predicted circRNAs

To validate whether the differentially expressed circRNAs found by sequencing are authentic, we selected exonic circRNAs that may have a significant differential expression for qRT-PCR verification. From these predicted circRNAs, six circRNAs of different abundance and lengths were selected. The six host genes are *PIWIL2*, *CATSPERD*, *SPATA6*, *ITPR2*, *CAMK2D*, *EGFR.* Primers spanning the backsplicing junction used in quantitative real-time PCR are shown in Appendix A. Figure 3 clearly shows that expression patterns of these selected circRNAs were concordant with RNA-seq results. These results demonstrated that the ribo-depleted total RNA-seq results were reliable.

## 4. Discussion

In this study, we performed the first demonstration of circNRA expression in donkey testis and caput epididymis. Using RNA-seq data, 19,721 circRNAs were identified, and we noted that almost all are exonic circRNAs. This characteristic was similar to previous observations on cattle, pig, and sheep testicular tissues [13].

Functional annotation of host genes found that multiple genes were annotated in spermatogenesis terms. Two circRNAs highly expressed in the testis were derived from the *PIWIL2* gene. *PIWIL2* belongs to the piwi family gene, a novel class of evolutionarily conserved genes [23]. The *PIWIL2* homolog *Miwi* has been identified in the mouse genome. *Miwi* is essential for male fertility and initiating spermiogenesis, a process that transforms round spermatids into mature sperm [24]. Four upregulated circRNAs from *CATSPERB* and *CATSPERD* genes. These two genes belonged to CatSper complex parts, were restricted to the testis, and were predicted to be active in the cilium. CatSperbeta (*CATSPERB*) is the first identified auxiliary protein to the CatSper channel [25]. Mice lacking the sperm tail-specific CATSPERD are infertile, and their spermatozoa lack both Ca^(2+)^ current and hyperactivated motility [26]. Spermatogenesis-associated (*SPATA*) genes are a large family of genes that play a vital role in testis development and spermatogenesis. Six circRNAs highly expressed in the testis were from SPATA genes, including *SPATA1*, *SPATA6*, *SPATA7*, and *SPATA22*. In mice, inactivation of *SPATA6* leads to acephalic spermatozoa and male sterility [27].

For the host genes of circRNAs highly expressed in caput epididymis, we found a particular TGF-β signaling pathway in KEGG pathway annotations. In total, 14 circRNAs from eight host genes were involved in this pathway, including transforming growth factor, β receptor Ⅱ (*TFGBR2*), latent transforming growth factor β binding protein 1 (*LTBP1*), activin A receptor type 2A (*ACVR2A*), and bone morphogenetic protein receptor (*BMPR*) family genes (*BMPR1A*, *BMPR1B*, and *BMPR2*). These genes’ progressive expression at key stages influences testis development and controls germline differentiation [28]. Our results enriched many significant signaling pathways related to reproductive regulation and neuroendocrine activity, including the GnRH signaling, estrogen, and calcium signaling pathways. Inositol 1,4,5-trisphosphate receptors (*ITPR*) are a family of intracellular Ca^2+^ release channels on the ER membrane. Calcium signaling plays essential roles in mammalian fertilization, such as the calcium wave in the egg and the acrosome reaction (AR) in sperm. The existence of ITPR types 1 and 3 in human sperm and their changes during the reaction, but ITPR type 2 was undetectable [29]. Our sequencing data only detected the host gene ITPR type 2 (*ITPR2*); this difference needs to be explored further. *ITPR1* and *ITPR2* play an essential and redundant role in maintaining the integrity of the fetal–maternal connection and embryonic viability [30]. In donkey’s caput epididymidis, the high expression of circRNAs may influence reproductive endocrine activity.

circRNAs were identified as efficient microRNA sponges [6]. Interestingly, the interactions of circRNAs and microRNAs show a peculiar behavior: multiple DE-circRNAs act on the same unique microRNA. This may be a redundant mechanism to regulate the expression of that microRNA. In our results, eca-miR-100 and eca-miR-205 were DEMs, which interacted with DECs. Previous studies have reported that miR-205 and miR-100 were explicitly expressed in human testis [31]. Among the target microRNAs of DECs, mir-532 and mir-34 families were also shown to be involved in cattle spermatogenesis [32,33]. MiR-122 is enriched in mice’s late-stage male germ cells, reducing mRNA expression of the post-transcriptionally regulated germ cell transition protein 2 (*Tnp2*) to affect male fertility [34,35]. Mir-122 was highly expressed in donkey testis, and there were many interacted circRNAs, including those derived from the *SPATA7* gene but no circRNAs derived from the *Tnp2* gene. It demonstrated that microRNA could act on multiple target genes to perform a function. Another microRNA, miR-184, was highly expressed in donkey testis and interacted with DECs from the PIWIL2 gene. Wu et al. reported that miR-184 expression levels increased with the postnatal development of the mouse testis, and its expression was restricted to testicular germ cells [34,35]. In the future, more molecular experiments will be conducted to verify the regulation of circRNAs on downstream miRNA targets.

## 5. Conclusions

In summary, we detected abundant circRNAs in donkey testes and epididymis and revealed the length distribution, genomic features, and other characteristics of circRNAs. Among these circRNAs from testis, 39 upregulated circRNA host genes were annotated in spermatogenesis terms, including *PIWIL2*, *CATSPERD*, *CATSPERB*, *SPATA6*, and *SYCP1*. Other host genes were annotated in the focal adhesion, Rap1 signaling pathway. Downregulated expressed circRNA host genes participated in the TGF-β signaling pathway, GnRH signaling pathway, estrogen signaling pathway, and calcium signaling pathway. These findings have laid the foundation for further study of noncoding RNA-related donkey reproduction.

## Figures and Tables

**Figure 1 genes-14-00066-f001:**
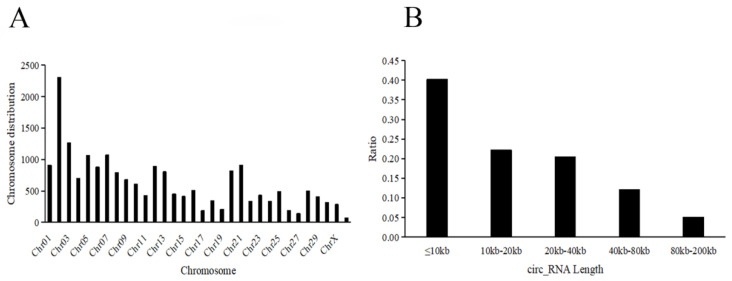
General characteristics of circRNAs in donkey testis and caput epididymis: (**A**) chromosomal distribution of circRNAs; (**B**) length distribution of circRNAs.

**Figure 2 genes-14-00066-f002:**
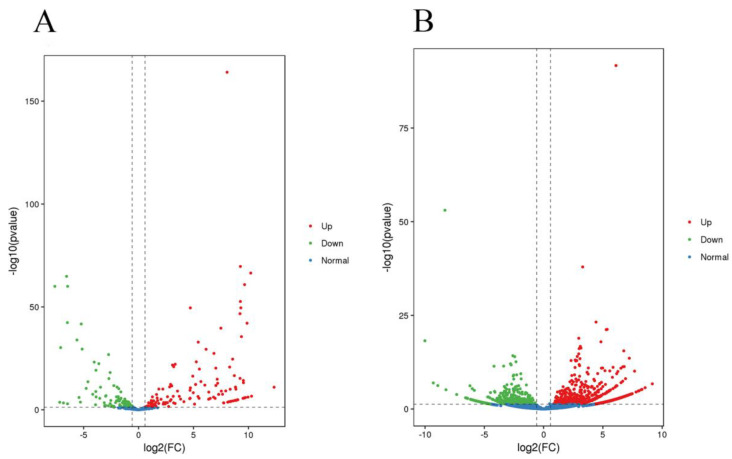
Differential microRNA (**A**) and circRNA (**B**) expression between testis and caput epididymis. Volcano plots showing −log_10_ (*p*-value) versus log_2_ (fold change) in RPM. Red dots denote significantly upregulated microRNAs and circRNAs, whereas green dots denote significantly downregulated microRNAs and circRNAs.

**Figure 3 genes-14-00066-f003:**
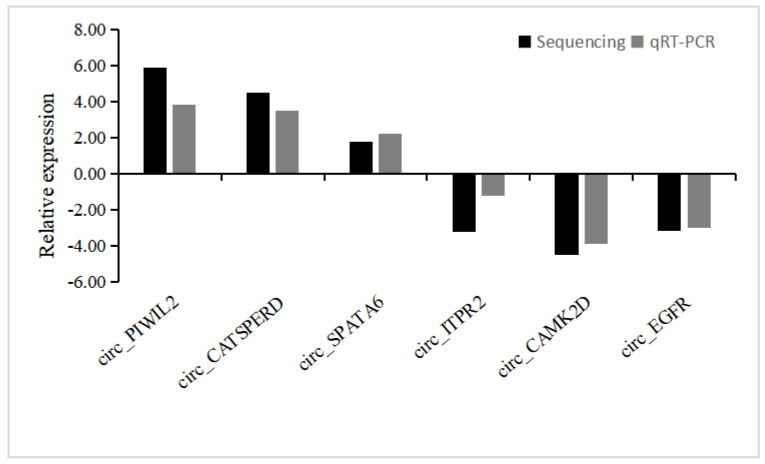
Validation of DE-circRNAs by qRT-PCR. Histograms of the relative expression levels. The y-axis shows the fold change between two groups (log(−ΔΔCt,2) for qRT-PCR, log_2_ (fold change) for sequencing). In total, the qRT-PCR results were consistent with the RNA-seq results.

## Data Availability

The data presented in this study are available on request from the corresponding author.

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
