# Peer review of "Identification of Circular RNAs of Testis and Caput Epididymis and Prediction of Their Potential Functional Roles in Donkeys"

_genes, 2022, doi:10.3390/genes14010066_

Round 1
Reviewer 1 Report
Dear authors,
The manuscript is interesting and add some new findings to the science. Please pay attention when constructing the final manuscript and the formatting of the document.
Line 136: Please check this citation "(Zhang er al., 2016)." all the other citations are numbered.
Author Response
A: Thanks for your suggestions. We have already corrected this error.
Reviewer 2 Report
This is an original paper focused on identifying circular RNAs of testis and caput epididymis and predicting their potential functional roles in donkeys. The results of the obtained research were properly described and discussed and brings some new knowledge.
Author Response
A: Thanks for your comments.
Reviewer 3 Report
Current study aimed to unravel the circRNAs and miRNAs specifically produced in testis and caput epididymis of Dezhou donkey. Furthermore, enriched pathways and biological processes asssociated with those circRNAs and miRNAs were given. Also statistical interactions between those circRNAs and miRNAs were suggested. The topic of the study is interesting and the study design is valid as far as understood. Therefore I would like to thank to the authors of the study for implementing such a valuable research. However, the English of the manuscript requires a comprehensive revision by a native/professional speaker, since the manuscript deeply suffer from clarity and coherency as well as spelling errors, mis-used words and lack of collocation which significantly defects fluency and understanding. Apart from that certain suggestions are given below:
- Line 33: Do you mean morphological assessment?
- Line 96: What are the criteria for 'clean data'? Can you please provide a little more detail on QC criteria with the relevant references.
- Line 100: Can you please detail the criteria for assembly and the quantitative analysis mentioned here.
- Line 102-103: It looks nice to see the implemented algorithm here. So, can you please approach the same for all the other software used in the M&M section. Because if you only provide the name of the software for the analyses, one cannot understand which algorithm, criteria, approach and model used for the analyses. That causes problems with the reproducability of the results.
- Can you please briefly give details of the analysis methods and models for especially statistical part (e.g., DEC and DEM detection etc.).
- Can you please provide details on how did you detect the targeted miRNA's of DECs.
- Certain parts of the text does not belong to the section given. For example, part of the results where the criteria for circRNA calling is given (line 156), in fact, belongs to the Material & Methods. Can you please carefully scan the text for those type of errors.
- Line 160: There is a part mentioning cauda epidydimis, which is irrelevant considering the study conducted.
-Line 186 and 190-191 also belong to the M&M section.
-Line 231-232 belongs to Introduction.
-Line 243-244: This type of statements belong to the discussion. Nonetheless, the aim and methodology of this study in not biologically validate this relationship but to suggest a statistical correlation, which might or might not lead to that conclusion. Your results are only an indicator of statistical relationships between certain circRNAs and miRNAs and does not validate the role of circRNAs as miRNA sponges. Therefore, I would not recommend presenting this statement as a result of such a study.
-There are repetitions in the manuscript such as those between lines 264-267. Such statements are usually given in M&M or Results section and usually once. The manuscript is suggested to be revised considering the removal of such repeated statements in different sections.
-Line 293-295: Not understandable.
- Even though the title of the manuscript is concerning circRNA's, the aim of the study and M&M is also concerned with the identification of differentially expressed miRNA's and the statistical relationships between DECs and DEMs. However, discussion section significantly lack the discussion of results concerning differentially expressed miRNA's per se and the importance of potential pathways/genes that is affected by the correlated circRNAs and miRNA's identified in this specific study. Extension of the manuscript regarding those is suggested.
- There is not any seperate Conclusion section in the manuscript. For the sake of a clear statement of what is concluded by this valuable study, it is suggested to point out a seperate Conclusion section with clear, evidence supported conclusions obtained in this study.
Author Response
We have uploaded a document file to respond reviewers questions.
